# Academic Tracker: Software for tracking and reporting publications associated with authors and grants

**P. Travis Thompson[1], Christian D. Powell[1,2,3], Hunter N. B. Moseley**[1,3,4,5,6]*

**1** Superfund Research Center, University of Kentucky, Lexington, KY, United States of America, **2** Department of Computer Science (Data Science Program), University of Kentucky, Lexington, KY, United States of America, **3** Markey Cancer Center, University of Kentucky, Lexington, KY, United States of America, **4** Department of Molecular and Cellular Biochemistry, University of Kentucky, Lexington, KY, United States of America, **5** Institute for Biomedical Informatics, University of Kentucky, Lexington, KY, United States of America, **6** Center for Clinical and Translational Science, University of Kentucky, Lexington, KY, United States of America

\* hunter.moseley@uky.edu

**Data Availability Statement:** All relevant data are located at: https://doi.org/10.6084/m9.figshare.19412165.

**Funding:** This work was supported in part by grants NSF 2020026 (PI Moseley - HNBM), NIH

## Abstract

In recent years, United States federal funding agencies, including the National Institutes of Health (NIH) and the National Science Foundation (NSF), have implemented public access policies to make research supported by funding from these federal agencies freely available to the public. Enforcement is primarily through annual and final reports submitted to these funding agencies, where all peer-reviewed publications must be registered through the appropriate mechanism as required by the specific federal funding agency. Unreported and/or incorrectly reported papers can result in delayed acceptance of annual and final reports and even funding delays for current and new research grants. So, it's important to make sure every peer-reviewed publication is reported properly and in a timely manner. For large collaborative research efforts, the tracking and proper registration of peer-reviewed publications along with generation of accurate annual and final reports can create a large administrative burden. With large collaborative teams, it is easy for these administrative tasks to be overlooked, forgotten, or lost in the shuffle. In order to help with this reporting burden, we have developed the Academic Tracker software package, implemented in the Python 3 programming language and supporting Linux, Windows, and Mac operating systems. Academic Tracker helps with publication tracking and reporting by comprehensively searching major peer-reviewed publication tracking web portals, including PubMed, Crossref, ORCID, and Google Scholar, given a list of authors. Academic Tracker provides highly customizable reporting templates so information about the resulting publications is easily transformed into appropriate formats for tracking and reporting purposes. The source code and extensive documentation is hosted on GitHub (https://moseleybioinformaticslab.github.io/academic_tracker/) and is also available on the Python Package Index (https://pypi.org/project/academic_tracker) for easy installation.

P42 ES007380 (PI Pennell; co-I HNBM) via the Data Management and Analysis Core (DMAC), and NIH U54 TR001998-05A1 (PI Kern; co-I HNBM). There was no additional external funding received for this study. The funders had no role in study design, data collection and analysis, decision to publish, or preparation of the manuscript.

**Competing interests:** The authors have declared that no competing interests exist.

## Introduction

Since 2008, the United States government has passed laws and issued directives to promote public access to peer-reviewed publications resulting from federal funding. These requirements started with Division G, Title II Section 218 of the Public Law (PL) 110–161 also known as the Consolidated Appropriations Act of 2008 [1], which directed the National Institutes for Health (NIH) to require all peer-reviewed publications supported by NIH funds to be electronically submitted to PubMed [2] within 12 months of the official date of publication [3]. Second in 2013, the White House Office of Science & Technology Policy (OSTP) mandated that all federal agencies with research and development budgets over $100 million to develop public access plans for research publications and data resulting from grants provided by these federal agencies [4]. Shortly thereafter in 2014, the US Congress passed the FY 2014 Omnibus Appropriations Act [5], which required federal agencies under Labor, Health and Human Services, and Education with research budgets of $100 million or more to provide public online access to peer-reviewed publications within 12 months of the official data of publication [6]. To comply with federal law, both NIH and NSF have implemented public access policies to make research supported by funding from these federal agencies freely available to the public. The enforcement of these policies typically occurs during the submission of annual and final reporting process for funded grants from NIH and NSF. In these reports, all peer-reviewed publications must be registered through the required mechanism by the specific federal funding agency. For NIH, peer-reviewed publications must be registered with PubMed Central and have a PubMed Central ID (PMCID). For NSF, peer-reviewed publications must be submitted to the NSF Public Access Repository (NSF-PAR) via Research.gov in the form of an archival PDF (PDF/A) [7]. Unreported and/or incorrectly reported papers can result in delayed acceptance of annual and final reports and funding delays for current and new research grants. Therefore, timely reporting of every peer-reviewed publication is required. For large collaborative research efforts involving large research teams or even multiple research teams, the tracking and proper registration of peer-reviewed publications along with generation of accurate annual and final reports can create a large administrative burden. With large collaborative teams, it is easy for these administrative tasks to be overlooked, forgotten, or lost in the shuffle.

In an effort to help researchers and their minders stay up-to-date with the reporting of peer-reviewed publications, we created the Academic Tracker software package. Written in the Python 3 programming language, Academic Tracker comprehensively searches major peer-reviewed publication tracking web portals, gathering relevant publications and useful tracking characteristics, for example, an indication of whether the publication has been reported to the NIH (is on PubMed), needs to be reported (is associated with an NIH grant), or satisfies the NIH's requirements to have a PMCID. It has the ability to search PubMed [2], ORCID [8], Google Scholar [9], and Crossref [10], given a list of authors and/or author IDs. Academic Tracker provides highly customizable reporting templates so information about the resulting publications is easily transformed into appropriate formats for tracking and reporting purposes.

ORCID (Open Researcher and Contributor ID) is a non-profit organization dedicated to uniquely identifying individuals who participate in research [8]. Once an author is registered, ORCID provides a unique ID that can be used to associate an author with their publications. These associations can be easily accessed from the ORCID website or through their application programming interface (API). Google Scholar is a search engine for scholarly literature with similar API search facilities to ORCID [9]. Authors can create profiles on Google Scholar, which Google Scholar uses to automatically associate publications with. Crossref is a non-profit association with both commercial and non-profit publisher members with a primary

purpose of enabling cross-publishing citation linking [10]. Crossref's stated goal is to make "research objects easy to find, cite, link, assess, and reuse." For the purposes of Academic Tracker, Crossref serves as a database with an easily accessible API for finding relevant publications.

Academic Tracker has three main use-cases and one supportive use-case. The first main use-case searches the aforementioned web portals for publications, given a list of authors. The second main use-case searches PubMed and Crossref for publication information, given a list of publication citations. Neither ORDID nor Google Scholar can be searched for specific publication information directly. ORCID is organized around author profiles and not publications themselves and does not provide a search option by publication characteristics. Google Scholar cannot be searched by specific publication characteristics, because Google Scholar has limited the repetitive programmatic use of their web service in this way. However, Google Scholar does allow repetitive programmatic search by author profile ID. The third main use-case finds collaborators given a list of authors. This is similar to the first use-case, but focuses on compiling the co-authors from the publications rather than the publications themselves. The fourth supportive use-case searches ORCID or Google Scholar for authors' unique IDs for these sources, given a list of authors.

The main output from the three main use-cases is a Javascript Object Notation (JSON) file containing information about each publication found. Other outputs vary on user settings. Customizable summary and project reports can be created with an option of emailing them as attachments. The collaborator report of the third use-case is also customizable. All emails are also copied into a JSON file. A configuration JSON file is needed as part of the input to Academic Tracker and the fourth supportive use-case will update this file with the information found during the search. A use-case diagram for Academic Tracker is shown in Fig 1.

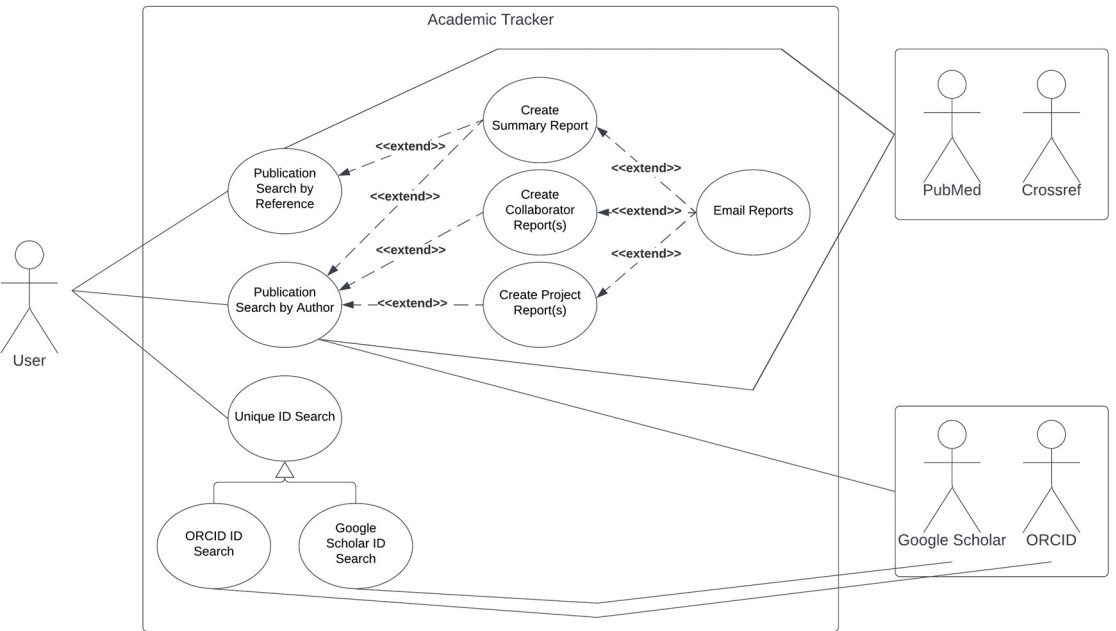

**Fig 1. Academic Tracker use-case diagram.** The first and third use-cases, publication search and collaborator search, are illustrated via the "Publication Search by Author" option. The second use-case, publication information, is illustrated via the "Publication Search by Reference" option. The supporting use-case, ORCID ID and Google Scholar ID searches, are illustrated by the "Unique ID Search" option.

## Methods

### 3<sup>rd</sup> party packages

Academic Tracker leverages many third-party Python libraries and packages to accomplish its major tasks. Academic Tracker uses the docopt library to implement a command line interface (CLI) from a Python docstring description. Next, Academic Tracker uses the jsonschema library to validate user JSON input against an expected schema, which is also in JSON format. JSON Schema is an independently developed vocabulary or framework created for the purpose of validating and annotating JSON. Other developers have implemented the vocabulary in several languages, and the jsonschema library is the Python language implementation. The specific schema used in Academic Tracker are in the Validation_Schemas directory of the supplemental materials. Academic Tracker also uses four different packages to query data sources for publications. Specifically, Academic Tracker uses the pymed, habanero, orcid, and scholarly libraries to query PubMed, Crossref, ORCID, and Google Scholar, respectively. For the second use-case, Academic Tracker uses the requests library to make HTTP requests and the beautifulsoup4 library to parse HTML in the pulled web pages given as the reference file. Next, Academic Tracker uses the fuzzywuzzy library to fuzzy match publication titles, which is necessary because publications do not have a universal unique identifier. For general file input/output, Academic Tracker uses several packages, including: i) the python-docx library to read Microsoft Word files, specifically for the reference file input; ii) the pandas library to read and write tabular data, specifically to read in author data and write out reports; and iii) indirectly the openpyxl library, which is used by pandas to write Excel files. In order to comprehensively compare publication information across different runs to see if any information has changed, Academic Tracker uses the deepdiff library. A list of packages and their versions are in Table 1.

### Use cases

Although there are 3 main use-cases and 1 supportive use-case, Academic Tracker has 2 main commands and 6 supporting commands (Table 2). The first and third main use-cases are handled by the author_search command, while the second main use-case is handled by the

**Table 1. Library dependencies for Academic Tracker.**

| Package | Version | Description | PyPI URL |
|---|---|---|---|
| docopt | 0.6.2 | Command line interface creation. | https://pypi.org/project/docopt/ |
| pymed | 0.8.9 | Query PubMed. | https://pypi.org/project/pymed/ |
| jsonschema | 3.0.1 | Validate JSON files. | https://pypi.org/project/jsonschema/ |
| habanero | 1.0.0 | Query Crossref. | https://pypi.org/project/habanero/ |
| orcid | 1.0.3 | Query ORCID. | https://pypi.org/project/orcid/ |
| scholarly | 1.4.5 | Query Google Scholar. | https://pypi.org/project/scholarly/ |
| beautifulsoup4 | 4.9.3 | HTML Parsing. | https://pypi.org/project/beautifulsoup4/ |
| fuzzywuzzy | 0.18.0 | Fuzzy match strings. | https://pypi.org/project/fuzzywuzzy/ |
| [a]python-docx | 0.8.11 | Read docx files. | https://pypi.org/project/python-docx/ |
| pandas | 0.24.2 | Read and write tabular files. | https://pypi.org/project/pandas/ |
| openpyxl | 2.6.2 | Used by pandas for writing Excel files. | https://pypi.org/project/openpyxl/ |
| requests | 2.21.0 | Make HTTP requests. | https://pypi.org/project/requests/ |
| deepdiff | 5.7.0 | Compare publication information. | https://pypi.org/project/deepdiff/ |

[a]The python-docx module imports with the name docx (original package name).

**Table 2. Description of Academic Tracker commands.**

| Command | Use-Case | Input Files | Output Files | Optional Files |
|---|---|---|---|---|
| author_search | 1 & 3 | Configuration JSON | Publications JSON | Summary Report Project Reports Collaborator Reports Emails JSON |
| reference_search | 2 | Configuration JSON Reference File | Publications JSON Tokenized Reference JSON | Summary Report Emails JSON |
| find_ORCID | S | Configuration JSON | Configuration JSON | |
| find_Google_Scholar | S | Configuration JSON | Configuration JSON | |
| add_authors | S | Configuration JSON Author CSV | Configuration JSON | |
| tokenize_reference | S | Reference File | Tokenized Reference JSON Tokenization Report | |
| gen_reports_and_emails_auth | S | Configuration JSON Publication JSON | | Summary Report Project Reports Collaborator Reports Emails JSON |
| gen_reports_and_emails_ref | S | Configuration JSON Reference File Publication JSON | Tokenized Reference JSON | Summary Report Emails JSON |

* "S" stands for supportive use case.

reference_search command. The supportive use-case is handled by the find_ORCID and find_Google_Scholar commands. The remaining four commands help users experiment with the tokenization and reporting systems in Academic Tracker and make it a little easier to convert author information into JSON format. The commands are listed in Table 2. The input and output files for each command are further described in Table 3.

## Module description

Although Academic Tracker is primarily designed to be a command line tool, it does provide an equivalent API, which can be utilized if so desired. The CLI and highest-level API for each command are implemented in the __main__.py file, but other submodules break down the steps into smaller pieces. Utilizing the API, reference_search and author_search are almost

**Table 3. Description of input and output file formats.**

| File | Format | Source | Description |
|---|---|---|---|
| config | JSON | Input/Output | Main input to direct the behavior of the program. Can also be an output for some commands. |
| publications | JSON | Input/Output | Main output from most commands that contains information about the publications found. Can also be an input for commands that generate reports. |
| reference | JSON URL docx txt MEDLINE | Input | Main input to search for publications by a reference. It can be an already tokenized reference JSON file, a URL to a webpage, a Word docx file, a txt file, or a MEDLINE formatted file. |
| emails | JSON | Output | A copy of any emails sent or would be sent if ran in test mode. |
| project_report | txt | Output | Depending on the settings in the config JSON file either a report summarizing information about publications found for the project or a report summarizing information about publications found for each author individually. |
| summary_report | txt | Output | A report summarizing information about all publications found for all projects and authors. |
| collaborators_report | csv | Output | A report summarizing information about co-authors for an author. |
| authors | csv | Input | A tabular csv file with authors to add or modify in the config JSON file. |
| tokenized_reference | JSON | Input/Output | A tokenized version of a reference. Can be an input to commands that work with reference sources. |

**Table 4. Submodules of Academic Tracker.**

| Submodule | Description |
|---|---|
| __main__.py | Contains the CLI and top-most API. |
| athr_srch_emails_and_reports.py | Contains functions for constructing the reports and emails. |
| athr_srch_modularized.py | Contains compartmentalized pieces of author_search. |
| athr_srch_webio.py | Contains the functions for querying the 4 data sources. |
| citation_parsing.py | Contains the functions for parsing and tokenizing reference sources. |
| fileio.py | Contains functions for reading and writing files. |
| helper_functions.py | Contains functions for common operations such as regex and data transformation. |
| ref_srch_emails_and_reports.py | Contains functions for constructing the reports and emails. |
| ref_srch_modularized.py | Contains compartmentalized pieces of author_search. |
| ref_srch_webio.py | Contains the functions for querying PubMed and Crossref, and some special case reference URLs. |
| tracker_schema.py | Contains the JSON schema used for validating user input. |
| user_input_checking.py | Contains functions for validating user input. |
| webio.py | Contains functions to interface with the internet. |

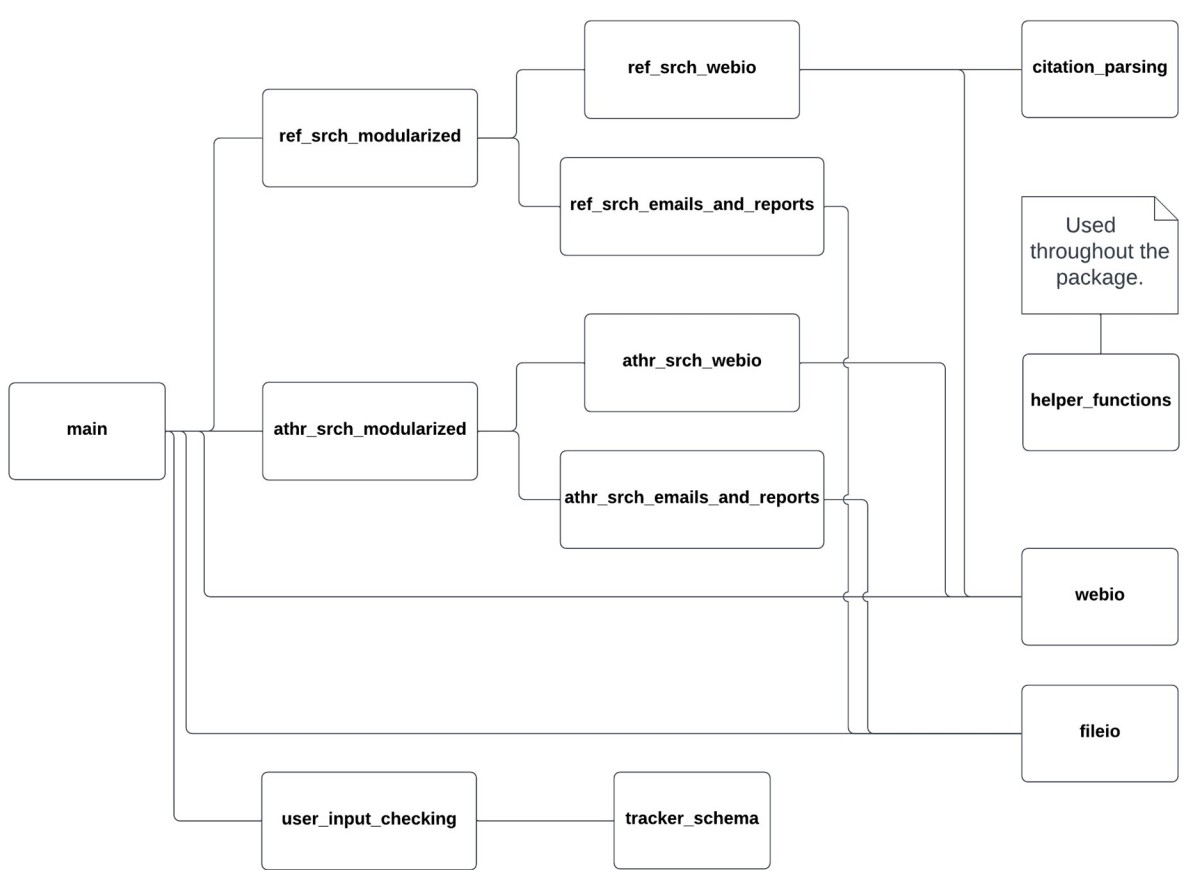

**Fig 2. Academic Tracker module diagram.** Submodule and module dependencies are illustrated by connecting lines, except for helper_functions which is utilized by most other submodules.

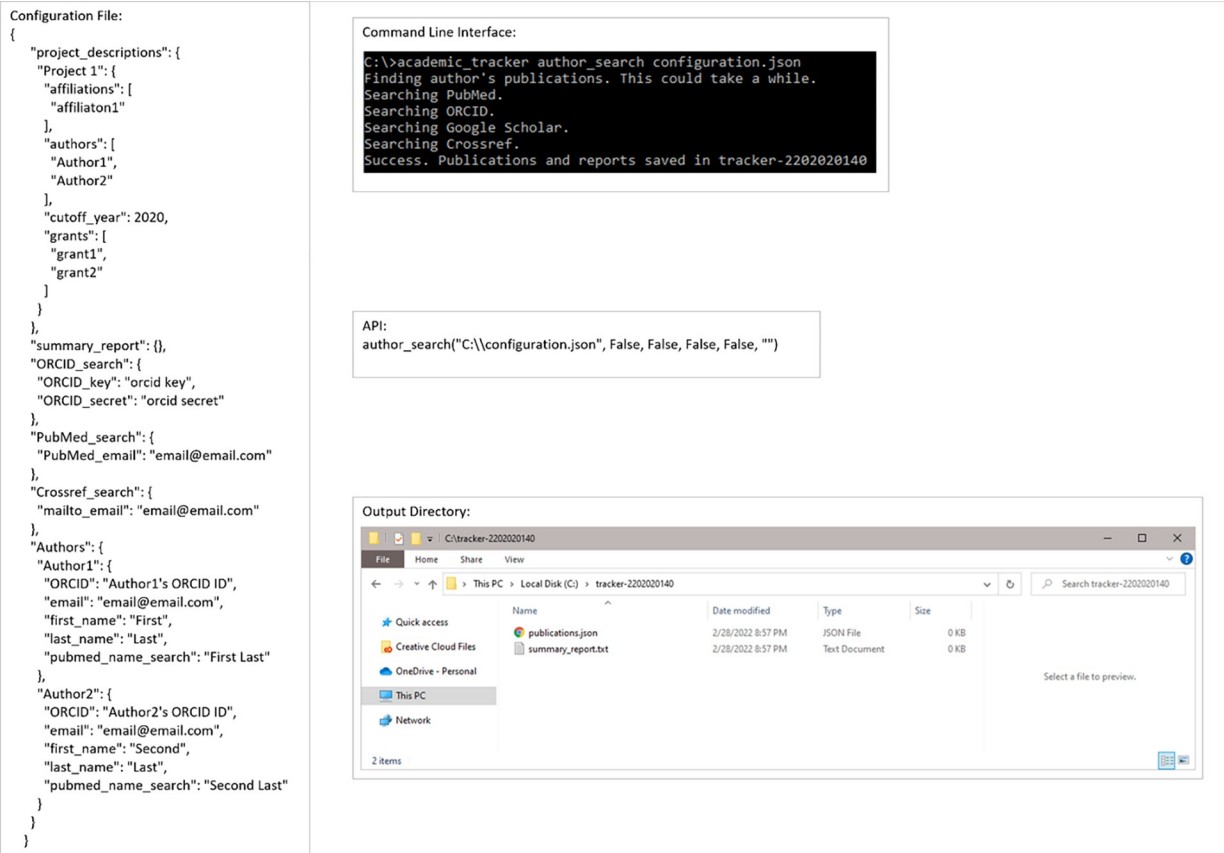

**Fig 3. Example execution of the author_search use-case.** Example configuration file, command-line execution, API execution, and file output of the author_search for publications use-case shown.

completely separated into their own submodules. The athr_srch_modularized.py submodule compartmentalizes the steps of author_search, while the athr_srch_webio.py and athr_srch_e-mails_and_reports.py submodules contain the functions to interface with the internet and generate reports and emails respectively. reference_search is organized the same way with the ref_srch_modularized.py, ref_srch_webio.py, and ref_srch_emails_and_reports.py submodules. The user_input_checking.py submodule contains the functions to validate user input for errors, and the tracker_schema.py submodule works in tandem with it to store the JSON schema being used for validation. The fileio.py submodule contains all the functions for reading and writing files. The webio.py submodule contains functions to interface with the internet that are more general purpose or common to multiple commands. It is where the functions to interface with the internet for find_ORCID and find_Google_Scholar are. The helper_functions.py submodule contains functions with common operations across all commands that don't classify well into any other submodule, such as regex operations and data transformation. The citation_parsing.py submodule contains all the functions used to tokenize the reference sources for reference_search. Table 4 lists the submodules of Academic Tracker, and Fig 2 shows a module diagram.

## Testing

The Academic Tracker package was originally developed in a Linux operating system (OS) environment, but has been directly tested on Linux, Windows, and MacOS operating systems.

Publications File:
```
{
  "32095784": {
    "PMCID": "PMC7039621",
    "abstract": null,
    "authors": [
      {
        "affiliation": "University of Kentucky College of Public Health.",
        "author_id": "Anna Hoover",
        "firstname": "Anna G",
        "initials": "AG",
        "lastname": "Hoover"
      },
      {
        "affiliation": "University of Kentucky Department of Dietetics and Human Nutrition.",
        "author_id": "Ann Koempel",
        "firstname": "Annie",
        "initials": "A",
        "lastname": "Koempel"
      },
      {
        "affiliation": "University of Kentucky College of Public Health.",
        "firstname": "W Jay",
        "initials": "WJ",
        "lastname": "Christian"
      },
      {
        "affiliation": "University of Kentucky College of Public Health.",
        "firstname": "Kimberly I",
        "initials": "KI",
        "lastname": "Tumlin"
      }
    ],
    "conclusions": null,
    "copyrights": null,
    "doi": null,
    "grants": [
      "G08 LM013185",
      "P30 ES026529",
      "P42 ES007380",
      "R01 ES032396"
    ],
    "journal": "Journal of Appalachian health",
    "keywords": [],
    "methods": null,
    "publication_date": {
      "day": 26,
      "month": 2,
      "year": 2020
    },
    "pubmed_id": "32095784",
    "results": null,
    "title": "Appalachian Environmental Health Literacy: Building Knowledge and Skills to Protect Health."
  },
  "http://abstracts.biomaterials.org/data/papers/2017/abstracts/0748.pdf": {
    "PMCID": null,
    "abstract": null,
    "authors": [
      {
        "affiliation": [
          "kentucky"
        ],
        "author_id": "Saiful Islam",
        "firstname": "Saiful",
        "initials": null,
        "lastname": "Islam"
      }
    ],
    "conclusions": null,
    "copyrights": null,
    "doi": null,
    "grants": null,
    "journal": null,
    "keywords": null,
    "methods": null,
    "publication_date": {
      "day": null,
      "month": null,
      "year": null
    },
    "pubmed_id": null,
    "results": null,
    "title": "Food-grade Zein Nanoparticles for Oral Delivery of Epigallocatechin-3-gallate (EGCG)"
  }
}
```

Summary Report:
```
Project 1:
  Author1:
    Title: Appalachian Environmental Health Literacy: Building Knowledge and Skills to Protect Health.
    Authors: Anna G Hoover, Annie Koempel, W Jay Christian, Kimberly I Tumlin
    Journal: Journal of Appalachian health
    DOI: None
    PMID: 32095784
    Grants: G08 LM013185, P30 ES026529, P42 ES007380, R01 ES032396

  Author2:
    Title: Food-grade Zein Nanoparticles for Oral Delivery of Epigallocatechin-3-gallate (EGCG)
    Authors: Saiful Islam
    Journal: None
    DOI: None
    PMID: None
    Grants: None Found
```

**Fig 4. Output file contents for the author_search use-case.** Example JSON publications output and plain-text summary report from the author_search for publications use-case shown.

All use-cases have been tested on these operating systems; however, Academic Tracker relies on sendmail or an emulator being installed and configured on the machine for its email functionality. In addition, each submodule includes unit-tests that test all critical functions of the submodule. Every function in every module is tested to make sure it gives the expected output when it should and errors when it should. All requests to web portals are replaced with mock data. The user_input_checking.py submodule has the largest number of tests, since it tests several error states for each element of the input JSON files. Every command line option is tested, for example, silent and not searching ORCID options. Various ways of creating reports are

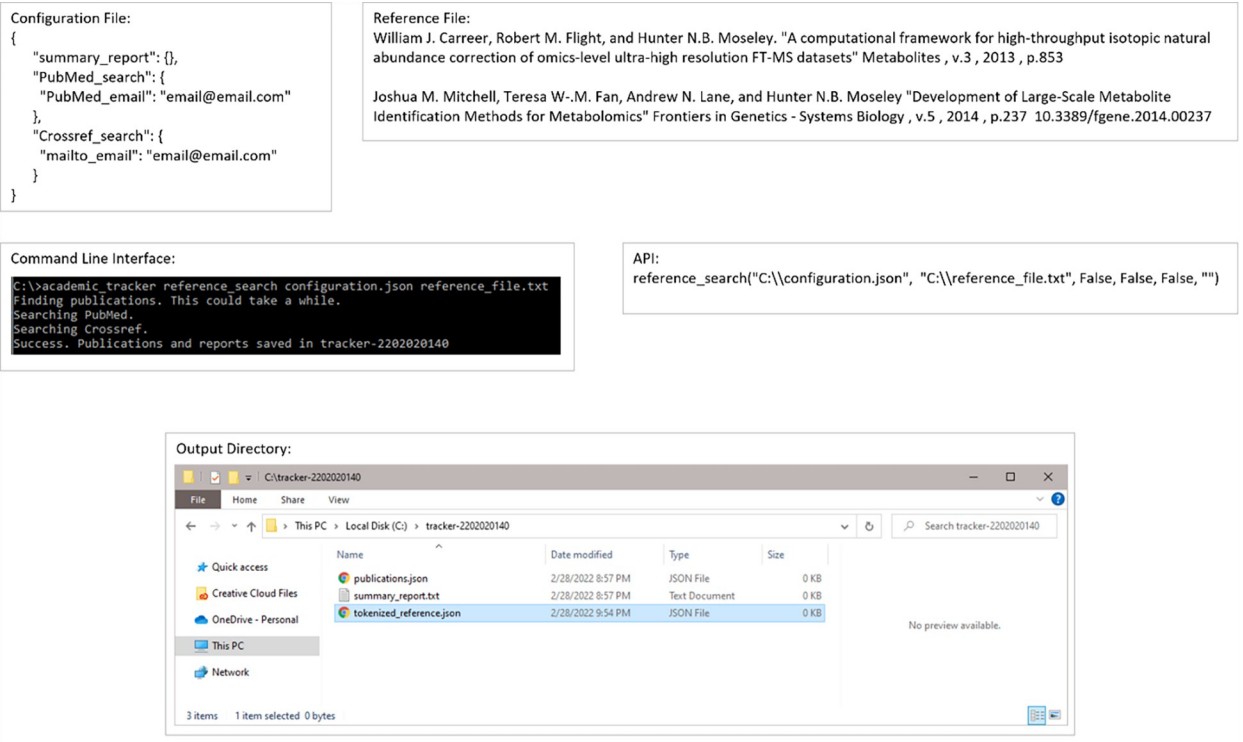

**Fig 5. Example execution of the reference_search use-case.** Example configuration file, reference file, command-line execution, API execution, and file output of the reference_search use-case shown.

also tested, such as creating a tabular report versus a text report, Excel versus CSV format, and renaming the report from the default name. Several different citation styles and sources are also tested to make sure they are tokenized correctly, such as MEDLINE, a MyNCBI bibliography URL, and an NSF Award page.

## Results

Academic Tracker can be utilized in many different ways and was designed with a great deal of flexibility, anticipating users' desire to use it in unpredictable ways. However, the three main and one supportive use-case are presented here. Note that the figures here are general examples with mostly dummy data. There are full examples with real data and run commands in the supplemental materials (Example_Runs subdirectory). The first main use-case involves searching for publications given author information. Fig 3 shows an example input configuration JSON file, the command line for its execution, the API execution equivalent, and the resulting output files. Fig 4 shows the contents of these resulting output files. Authors without unique ORCID or Google Scholar IDs are identified by matching first name, last name, and at least one affiliation.

The second main use-case involves looking for publications based on a given reference. Fig 5 shows an example input configuration JSON file, the command line for its execution, the API execution equivalent, and the resulting output files. Figs 6 and 7 show the contents of the resulting output files.

The third use-case is basically identical to the first, but a collaborator report attribute needs to be added to an author. Fig 8 is essentially the same as Fig 3, but with a collaborator report attribute added to Author1 and the report in the output directory. Fig 4 already shows the

Publications File:

```
{
"https://doi.org/10.3390/metabo3040853": {
  "PMCID": "PMC3882318",
  "abstract": null,
  "authors": [
    {
      "affiliation": "Department of Molecular and Cellular Biochemistry, University of Kentucky, Lexington, KY
40536, USA; jim.carreer@uky.edu (W.J.C.); robert.flight@uky.edu (R.M.F.).",
      "firstname": "William J",
      "initials": "WJ",
      "lastname": "Carreer"
    },
    {
      "affiliation": "Department of Molecular and Cellular Biochemistry, University of Kentucky, Lexington, KY
40536, USA; jim.carreer@uky.edu (W.J.C.); robert.flight@uky.edu (R.M.F.).",
      "firstname": "Robert M",
      "initials": "RM",
      "lastname": "Flight"
    },
    {
      "affiliation": "Department of Molecular and Cellular Biochemistry, University of Kentucky, Lexington, KY
40536, USA; jim.carreer@uky.edu (W.J.C.); robert.flight@uky.edu (R.M.F.).",
      "firstname": "Hunter N B",
      "initials": "HN",
      "lastname": "Moseley"
    }
  ],
  "conclusions": null,
  "copyrights": null,
  "doi": "10.3390/metabo3040853",
  "grants": [
    "P20 GM103436",
    "P20 RR016481",
    "R01 ES022191",
    "U24 DK097215"
  ],
  "journal": "Metabolites",
  "keywords": [
    "Fourier transform mass spectrometry",
    "analytical derivation",
    "multi-isotope natural abundance correction",
    "parallelization",
    "stable isotope tracing",
    "stable isotope-resolved metabolomics"
  ],
  "methods": null,
  "publication_date": {
    "day": 10,
    "month": 1,
    "year": 2014
  },
  "pubmed_id": "24404440",
  "results": null,
  "title": "A Computational Framework for High-Throughput Isotopic Natural Abundance Correction of Omics-
Level Ultra-High Resolution FT-MS Datasets."
},
```

```
"https://doi.org/10.1186/1471-2105-15-s10-p36": {
  "PMCID": null,
  "abstract": null,
  "authors": [
    {
      "affiliation": null,
      "firstname": "Joshua M",
      "initials": null,
      "lastname": "Mitchell"
    },
    {
      "affiliation": null,
      "firstname": "Teresa W-M",
      "initials": null,
      "lastname": "Fan"
    },
    {
      "affiliation": null,
      "firstname": "Andrew N",
      "initials": null,
      "lastname": "Lane"
    },
    {
      "affiliation": null,
      "firstname": "Hunter NB",
      "initials": null,
      "lastname": "Moseley"
    }
  ],
  "conclusions": null,
  "copyrights": null,
  "doi": "10.1186/1471-2105-15-s10-p36",
  "grants": null,
  "journal": "Springer Science and Business Media LLC",
  "keywords": null,
  "methods": null,
  "publication_date": {
    "day": null,
    "month": null,
    "year": 2014
  },
  "pubmed_id": null,
  "results": null,
  "title": "Development of large-scale metabolite identification methods for metabolomics"
  }
}
```

Tokenized Reference:

```
[
  {
    "DOI": null,
    "PMID": null,
    "authors": [
      {
        "first": "William",
        "last": "Carreer",
        "middle": "J."
      },
      {
        "first": "Robert",
        "last": "Flight",
        "middle": "M."
      },
      {
        "first": "Hunter",
        "last": "Moseley",
        "middle": "N.B."
      }
    ],
    "pub_dict_key": "",
    "reference_line": "William J. Carreer, Robert M. Flight, and Hunter N.B. Moseley. \"A computational
framework for high-throughput isotopic natural abundance correction of omics-level ultra-high resolution FT-MS
datasets\" Metabolites , v.3 , 2013 , p.853",
    "title": "A computational framework for high-throughput isotopic natural abundance correction of omics-level
ultra-high resolution FT-MS datasets"
  },
```

```
  {
    "DOI": null,
    "PMID": null,
    "authors": [
      {
        "first": "Joshua",
        "last": "Mitchell",
        "middle": "M."
      },
      {
        "first": "Teresa",
        "last": "Fan",
        "middle": "W-.M."
      },
      {
        "first": "Andrew",
        "last": "Lane",
        "middle": "N."
      },
      {
        "first": "Hunter",
        "last": "Moseley",
        "middle": "N.B."
      }
    ],
    "pub_dict_key": "",
    "reference_line": "Joshua M. Mitchell, Teresa W-.M. Fan, Andrew N. Lane, and Hunter N.B. Moseley
\"Development of Large-Scale Metabolite Identification Methods for Metabolomics\" Frontiers in Genetics -
Systems Biology , v.5 , 2014 , p.237  10.3389/fgene.2014.00237",
    "title": "Development of Large-Scale Metabolite Identification Methods for Metabolomics"
  }
]
```

**Fig 6. Publication JSON and tokenized reference contents for the reference_search use-case.**

Summary Report:
Reference Line: William J. Carreer, Robert M. Flight, and Hunter N.B. Moseley. \"A computational framework for high-throughput isotopic natural abundance correction of omics-level ultra-high resolution FT-MS datasets\" Metabolites , v.3 , 2013 , p.853
Tokenized Reference:
  Authors: William Carreer, Robert Flight, Hunter Moseley
  Title: A computational framework for high-throughput isotopic natural abundance correction of omics-level ultra-high resolution FT-MS datasets
  PMID: None
  DOI: None
Queried Information:
  DOI: 10.3390/metabo3040853
  PMID: 24404440
  PMCID: PMC3882318
  Grants: P20 GM103436, P20 RR016481, R01 ES022191, U24 DK097215

Reference Line: Joshua M. Mitchell, Teresa W-.M. Fan, Andrew N. Lane, and Hunter N.B. Moseley \"Development of Large-Scale Metabolite Identification Methods for Metabolomics\" Frontiers in Genetics - Systems Biology , v.5 , 2014 , p.237  10.3389/fgene.2014.00237
Tokenized Reference:
  Authors: Joshua Mitchell, Teresa Fan, Andrew Lane, Hunter Moseley
  Title: Development of Large-Scale Metabolite Identification Methods for Metabolomics
  PMID: None
  DOI: None
Queried Information:
  DOI: 10.1186/1471-2105-15-s10-p36
  PMID: None
  PMCID: None
  Grants: None

**Fig 7. Summary report contents for the reference_search use-case.**

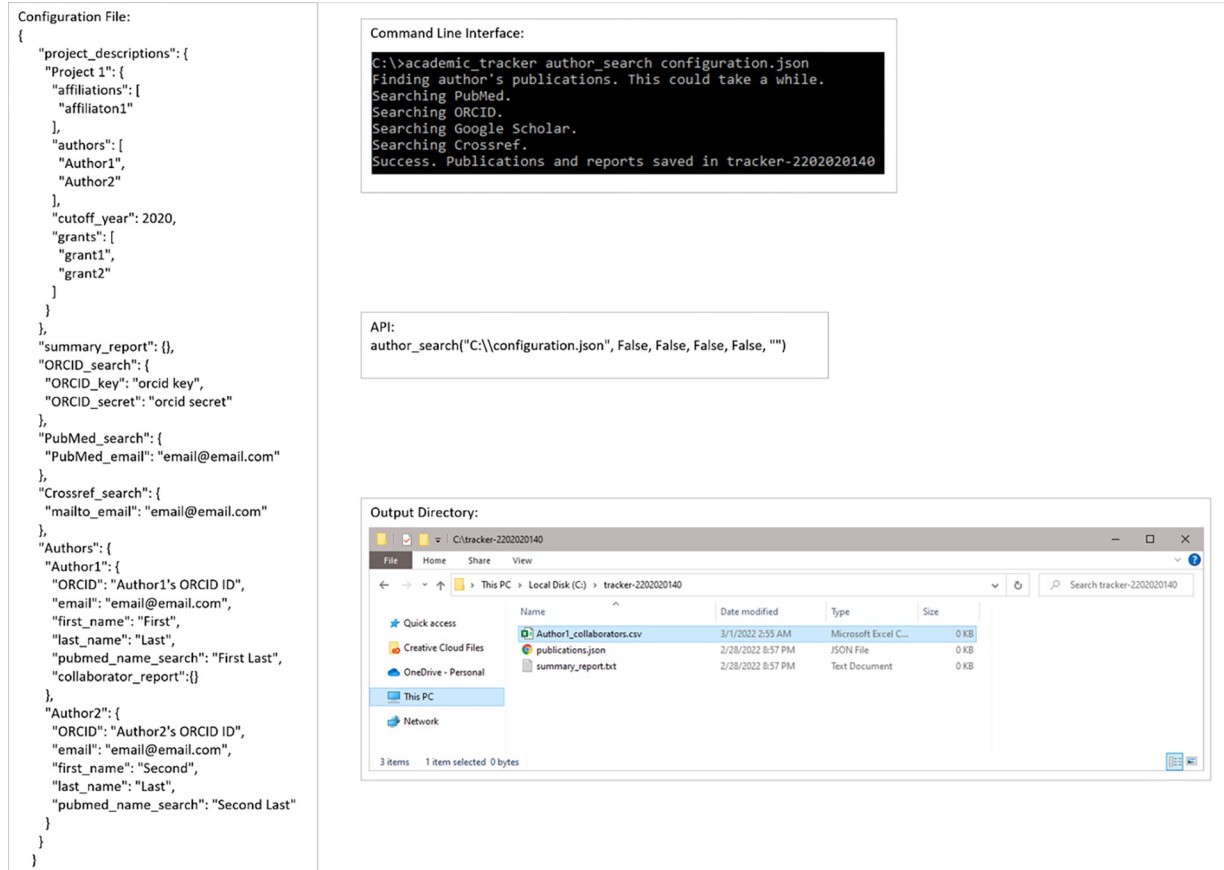

**Fig 8. Example execution of collaborator report generation use-case.** Example configuration file, command-line execution, API execution, and file output of the author_search for collaborators use-case shown.

**Table 5. Contents of the output collaborator report as a table.**

| Name | Affiliations |
| --- | --- |
| Christian, W Jay | University of Kentucky College of Public Health. |
| Hoover, Anna G | University of Kentucky College of Public Health. |
| Koempel, Annie | University of Kentucky Department of Dietetics and Human Nutrition. |
| Tumlin, Kimberly I | University of Kentucky College of Public Health. |

contents of the publications JSON and summary report. Table 5 shows the contents of the resulting collaborator report table.

The supportive use-case is broken into 2 commands: find_ORCID for finding ORCID IDs and find_Google_Scholar for finding Google Scholar IDs. Fig 9 shows an example input configuration JSON, how to accomplish this using the command line and API, and the resulting output files for finding ORCID IDs. Fig 10 shows the contents of the resulting configuration JSON file. Figs 11 and 12 are the same as Figs 9 and 10 but for finding Google Scholar IDs.

## Discussion and conclusions

Academic Tracker is a useful tool for querying major scientific publication web portals for publications, given a list of authors or references and for creating highly customizable reports from the list of publications found. The software package provides assistance in repetitive tracking and reporting of peer-reviewed publications associated with specific authors, projects, and grants. Specifically, the JSON configuration file supports batch execution, directing Academic Tracker to perform multiple related author searches and report generations. The JSON configuration file has many optional parameters to customize searching and report generation, including a cutoff_year for searching. Academic Tracker is also designed for repetitive tracking by comparing current search results to prior search results to limit reporting to changes in publications detected and in publication attributes. Academic Tracker also provides facilities for generating lists of co-author collaborators, which has several uses in grant proposal submission. But given the number of major use-cases and versality of the software, there is some intellectual overhead required to initially setup the JSON configuration file and customize reports. Additional supportive commands are included to make learning and troubleshooting the tool

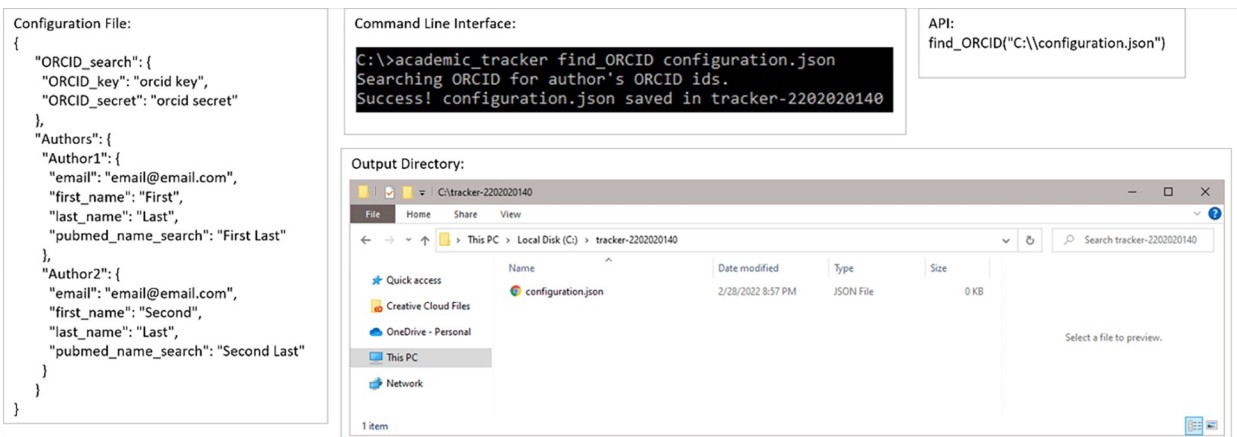

**Fig 9. Example execution of the ORCID ID search use-case.** Example configuration file, command-line execution, API execution, and file output of the author ORCID ID search use-case shown.

```
Configuration File:
{
    "ORCID_search": {
     "ORCID_key": "orcid key",
     "ORCID_secret": "orcid secret"
    },
    "Authors": {
     "Author1": {
       "ORCID": "Author1's ORCID ID",
       "email": "email@email.com",
       "first_name": "First",
       "last_name": "Last",
       "pubmed_name_search": "First Last"
     },
     "Author2": {
       "ORCID": "Author2's ORCID ID",
       "email": "email@email.com",
       "first_name": "Second",
       "last_name": "Last",
       "pubmed_name_search": "Second Last"
     }
```

**Fig 10. Contents of the output configuration JSON file for the ORCHID ID search use-case.**

easier for new users. Also, there is extensive documentation available to help with the learning curve: https://moseleybioinformaticslab.github.io/academic_tracker/

In addition, when installed via the Python package management system pip, a console script "academic_tracker" is created automatically for the user, providing easy access to the CLI.

While the package accesses multiple major peer-reviewed publication tracking web portals, it is fundamentally limited to the information provided by these web portals and must assume the information provided is accurate. One possibility is to download a PDF of the publication itself for analysis. However, this is pragmatically infeasible, since there is wide variation in how journals organize the splash page of their publications. One way to alleviate this issue is for

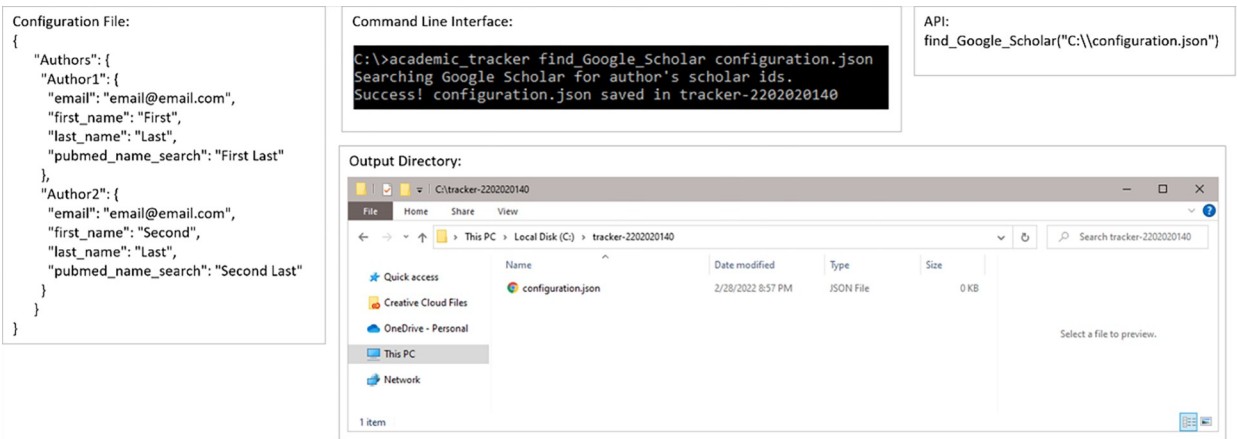

**Fig 11. Example execution of the Google Scholar ID search use-case.** Example configuration file, command-line execution, API execution, and file output of the author Google Scholar ID search use-case shown.

```
Configuration File:
{
    "Authors": {
     "Author1": {
       "scholar_id": "Author1's scholar ID",
       "email": "email@email.com",
       "first_name": "First",
       "last_name": "Last",
       "pubmed_name_search": "First Last"
     },
     "Author2": {
       "scholar_id": "Author2's scholar ID",
       "email": "email@email.com",
       "first_name": "Second",
       "last_name": "Last",
       "pubmed_name_search": "Second Last"
     }
    }
}
```

**Fig 12. Contents of the output configuration JSON file for the Google Scholar ID search use-case.**

journals to adopt a DOI extension like ".pdf" which would link directly to the PDF version of the publication, if the PDF version is accessible. This is similar to the versioning ".v#" DOI extension that FigShare uses to provide access each version of a public FigShare repository. If a practical way to directly access the PDF is implemented either by journals or the publication tracking web portals, we would extend Academic Tracker to utilize it. Still in its current implementation, we believe Academic Tracker can significantly reduce the stress and hassle of reporting publications to federal funding agencies, reducing the chance for accidental non-compliance and resulting delay in funding.

## Acknowledgments

We also thank Jennifer Moore for feedback during the development of the report generation capabilities.

## Author Contributions

**Conceptualization:** Christian D. Powell, Hunter N. B. Moseley.

**Funding acquisition:** Hunter N. B. Moseley.

**Methodology:** P. Travis Thompson, Christian D. Powell, Hunter N. B. Moseley.

**Project administration:** Hunter N. B. Moseley.

**Software:** P. Travis Thompson.

**Supervision:** Hunter N. B. Moseley.

**Validation:** P. Travis Thompson.

**Writing – original draft:** P. Travis Thompson, Hunter N. B. Moseley.

**Writing – review & editing:** P. Travis Thompson, Christian D. Powell, Hunter N. B. Moseley.

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
