## [Decision Letter · Decision Letter 0]

22 Jul 2022

PONE-D-22-09700Academic Tracker: Software for Tracking and Reporting Publications Associated with Authors and GrantsPLOS ONE

Dear Dr. Moseley,

Thank you for submitting your manuscript to PLOS ONE. After careful consideration, we feel that it has merit but does not fully meet PLOS ONE’s publication criteria as it currently stands. Therefore, we invite you to submit a revised version of the manuscript that addresses the points raised during the review process.

We look forward to receiving your revised manuscript.

Kind regards,

Yuji Zhang

Academic Editor

PLOS ONE

Journal Requirements:

This work was supported in part by grants NSF 2020026 (PI Moseley - HNBM), NIH P42 ES007380 (PI Pennell; co-I HNBM) via the Data Management and Analysis Core (DMAC), and NIH U54 TR001998-05A1 (PI Kern; co-I HNBM).

This work was supported in part by grants NSF 2020026 (PI Moseley - HNBM), NIH P42 ES007380 (PI Pennell; co-I HNBM) via the Data Management and Analysis Core (DMAC), and NIH U54 TR001998-05A1 (PI Kern; co-I HNBM).

This work was supported in part by grants NSF 2020026 (PI Moseley), NIH P42 ES007380 (PI Pennell) via the Data Management and Analysis Core (DMAC), and NIH U54 TR001998-05A1 (PI Kern). 

However, funding information should not appear in the Acknowledgments section or other areas of your manuscript. We will only publish funding information present in the Funding Statement section of the online submission form. 

This work was supported in part by grants NSF 2020026 (PI Moseley - HNBM), NIH P42 ES007380 (PI Pennell; co-I HNBM) via the Data Management and Analysis Core (DMAC), and NIH U54 TR001998-05A1 (PI Kern; co-I HNBM).

Reviewers' comments:

Reviewer's Responses to Questions

**Comments to the Author**

1. Is the manuscript technically sound, and do the data support the conclusions?

Reviewer #1: Yes

Reviewer #2: Yes

2. Has the statistical analysis been performed appropriately and rigorously? 

Reviewer #1: N/A

Reviewer #2: N/A

3. Have the authors made all data underlying the findings in their manuscript fully available?

Reviewer #1: Yes

Reviewer #2: Yes

4. Is the manuscript presented in an intelligible fashion and written in standard English?

Reviewer #1: Yes

Reviewer #2: Yes

5. Review Comments to the Author

Reviewer #1: Thompson et al. developed a python package called “Academic Tracker”, which searches public repositories for federally funded publications linked to an author or a list of authors. The authors demonstrated its applications through numerous use-cases. Overall, the manuscript is well written and is accompanied by a nice online documentation and tutorial. The supplemental data include real examples to help running the scripts. The tool is certainly useful for the search community.

Major points:

1). The authors may consider adding subtitles in the "Methods" part, which will help the readers to follow the design and built-in functions more easily.

2). Table 4 lists submodules in the package, which are also illustrated in Fig. 2. It is less straightforward to link Table 4 to Fig. 2. Consider adding an extra column in Table 4 to assign a number (1..13) to each of the 13 submodules, and then adding this number to Figure 2, something like "1: main" and "13: citation_parsing.py", OR "main (1)" and "citation_parsing.py (13)" in Fig.2.

3). Possible limitations or future enhancement of the package need to be discussed.

Minor points:

1). In Table 1, the docx package v0.2.4 was not found, please double check.

2). In Table 2, what is "S" stands for? supporting commands?

3). It is less clear what citation format is required for a reference file, related to Reference File shown in Fig.5

4). page 22, "One for finding ORCID IDs and the other for finding Google Scholar IDs"

better to be more specific, e.g., One (find_ORCID submodule) for finding ORCID IDs and the other (find_Google_Scholar submodule) for finding Google Scholar IDs".

Reviewer #2: Have you looked into scopus (https://www.scopus.com/freelookup/form/author.uri)? Can you comment why it is not included in your software?

Why don't you use public SMTP server such as gmail to simplify the local installation and setup of your software?

How do you handle author reconciliation? i. e. different authors may have the same name, how could you tell the one you search function returns is the right one?

Can you look into the actual report formats of the federal funding agencies', for example RPPR from NIH, so your tool can create the report ready to be used for such reporting tools.

People also use different affiliations (sometime, different spelling of affiliations) in their publications, how do you handle this?

For your search results, can you provide some statistics, such as precision and recall?

In your tutorials, you should provide some concrete config_file.json file instead of dummy file. So the user can run your example immediately to see its action instead of having to create the workable config file first. You should also provide a corresponding report files, so the user can know what they would expect.

6. PLOS authors have the option to publish the peer review history of their article (what does this mean?). If published, this will include your full peer review and any attached files.

Reviewer #1: No

Reviewer #2: No

---

## [Author Response · Author response to Decision Letter 0]

31 Jul 2022

Reviewer #1 

Thompson et al. developed a python package called “Academic Tracker”, which searches public repositories for federally funded publications linked to an author or a list of authors. The authors demonstrated its applications through numerous use-cases. Overall, the manuscript is well written and is accompanied by a nice online documentation and tutorial. The supplemental data include real examples to help running the scripts. The tool is certainly useful for the search community.

Response:

We thank the reviewer for their positive comments about the utility of the package and its documentation. We strive to reach the highest industrial coding quality and practices in our public codebases.

Issue 1:

Major points:

1). The authors may consider adding subtitles in the "Methods" part, which will help the readers to follow the design and built-in functions more easily.

Response:

This is an excellent suggestion! We have added subtitles.

Issue 2:

2). Table 4 lists submodules in the package, which are also illustrated in Fig. 2. It is less straightforward to link Table 4 to Fig. 2. Consider adding an extra column in Table 4 to assign a number (1..13) to each of the 13 submodules, and then adding this number to Figure 2, something like "1: main" and "13: citation_parsing.py", OR "main (1)" and "citation_parsing.py (13)" in Fig.2.

Response:

This is an interesting suggestion, but numbers do not make much sense in Figure 2. But we put Table 4 in alphabetical order. This should make it easier for the reader to cross-reference Figure 2 with Table 4.

Issue 3:

3). Possible limitations or future enhancement of the package need to be discussed.

Response:

We add the following to the discussion:

“While the package accesses multiple major peer-reviewed publication tracking web portals, it is fundamentally limited to the information provided by these web portals. One possibility is to download a PDF of the publication itself for analysis. However, this is pragmatically infeasible, since there is wide variation in how journals organize the splash page of their publications. One way to alleviate this issue is for journals to adopt a DOI extension like “.pdf” which would link directly to the PDF version of the publication, if the PDF version is accessible. This is similar to the versioning “.v#” DOI extension that FigShare uses to provide access each version of a public FigShare repository. If a practical way to directly access the PDF is implemented either by journals or the publication tracking web portals, we would extend Academic Tracker to utilize it.”

Issue 4:

Minor points:

1). In Table 1, the docx package v0.2.4 was not found, please double check.

Response:

While the docx package exists and the last version was 0.2.4, it was moved into a new package called python-docx which is version 0.8.11. The link in Table 1 was correct, but the naming could be misleading, and the version was incorrect. It still imports in Python as “docx” not “python-docx”, which leads to confusion, so we added a footnote to Table 1 that highlights this fact. Of course, everything was correct in the program requirements files and the codebase itself.

Issue 5:

2). In Table 2, what is "S" stands for? supporting commands?

Response:

It stands for “supportive use-case”. We have added it as a footer note to the table. 

Issue 6:

3). It is less clear what citation format is required for a reference file, related to Reference File shown in Fig.5

Response:

This is described in Tutorial > Search For Publications By Reference (https://moseleybioinformaticslab.github.io/academic_tracker/tutorial.html#search-for-publications-by-reference) and Tokenization (https://moseleybioinformaticslab.github.io/academic_tracker/tokenization.html) in the documentation. We decided that this was too much detail for the paper, which is meant to introduce the package and its functionality. We expect someone to read the documentation to learn how to actually use the package.

Issue 7:

4). page 22, "One for finding ORCID IDs and the other for finding Google Scholar IDs"

better to be more specific, e.g., One (find_ORCID submodule) for finding ORCID IDs and the other (find_Google_Scholar submodule) for finding Google Scholar IDs".

Response:

Excellent suggestion! We made this change, but in a slightly different way.

Reviewer #2

Issue 1:

Have you looked into scopus (https://www.scopus.com/freelookup/form/author.uri)? Can you comment why it is not included in your software?

Response:

Scopus is a product of Elsevier and has a paywall. The freelookup (Scopus Preview) has many limitations including only viewing an author’s last 10 documents. We only implemented access to free services.

Issue 2:

Why don't you use public SMTP server such as gmail to simplify the local installation and setup of your software?

Response:

Installation is already very easy via “pip install academic-tracker” which then makes upgrading very easy. Installation via an emailed or downloaded package does not enable easy upgrade to the latest version of the package.

Issue 3:

How do you handle author reconciliation? i. e. different authors may have the same name, how could you tell the one you search function returns is the right one?

Response:

This is why we include ORCID and Google Scholar profiles. Also, the package uses matching first name, last name, and at least one affiliation to separate authors with identical names. We have added these details to the manuscript:

“Authors without unique ORCID or Google Scholar IDs are identified by matching first name, last name, and at least one affiliation.”

Also, this is described in the package README under How Authors Are Identified (https://moseleybioinformaticslab.github.io/academic_tracker/index.html).

Issue 4:

Can you look into the actual report formats of the federal funding agencies', for example RPPR from NIH, so your tool can create the report ready to be used for such reporting tools.

Response:

The reporting format can be specified. We plan to expand the example_configs subdirectory to include additional reporting templates. 

https://github.com/MoseleyBioinformaticsLab/academic_tracker/tree/main/example_configs

Issue 5:

People also use different affiliations (sometime, different spelling of affiliations) in their publications, how do you handle this?

Response:

Multiple affiliations can be associated with a given author. This is needed, since an author may have been at multiple institutions over their career. But allowing a list in the configuration file allows for multiple spellings and abbreviations to be included. However, the best way to deal with these situations is for authors to register for ORCID IDs and Scholar Profile IDs.

Issue 6:

For your search results, can you provide some statistics, such as precision and recall?

Response:

This would require having a golden dataset or checking a lot of results by hand. Neither is easy to obtain. Also, how would we handle errors or even differences in the publication tracking web portals. We have reasonably unit-tested the codebase and tested the package in both development and practical use for the University of Kentucky Superfund Research Center. We have discovered that discrepancies between query results provided by publication tracking web portals sometimes leads to duplicate publication entries in Academic Tracker results, but at least it includes everything in the report. We have added this limitation of relying on the publication tracking web portals into the Discussion section:

“While the package accesses multiple major peer-reviewed publication tracking web portals, it is fundamentally limited to the information provided by these web portals and must assume the information provided is accurate. One possibility is to download a PDF of the publication itself for analysis. However, this is pragmatically infeasible, since there is wide variation in how journals organize the splash page of their publications. One way to alleviate this issue is for journals to adopt a DOI extension like “.pdf” which would link directly to the PDF version of the publication, if the PDF version is accessible. This is similar to the versioning “.v#” DOI extension that FigShare uses to provide access each version of a public FigShare repository. If a practical way to directly access the PDF is implemented either by journals or the publication tracking web portals, we would extend Academic Tracker to utilize it.”

Issue 7:

In your tutorials, you should provide some concrete config_file.json file instead of dummy file. So the user can run your example immediately to see its action instead of having to create the workable config file first. You should also provide a corresponding report files, so the user can know what they would expect.

Response:

We already have concrete examples and corresponding report files in the supplemental material. We also have concrete example config files in the example_configs folder of the package:

https://github.com/MoseleyBioinformaticsLab/academic_tracker/tree/main/example_configs

So we do not see any additional utility in adding them to the tutorial. Also, it would be problematic to maintain their corresponding report files in the tutorial, since these will change over time, unless we created concrete examples involved deceased scientists. Even then, a paper or two have been known to be published posthumously.

---

## [Decision Letter · Decision Letter 1]

4 Nov 2022

Academic Tracker: Software for Tracking and Reporting Publications Associated with Authors and Grants

PONE-D-22-09700R1

Dear Dr. Moseley,

We’re pleased to inform you that your manuscript has been judged scientifically suitable for publication and will be formally accepted for publication once it meets all outstanding technical requirements.

Kind regards,

Yuji Zhang

Academic Editor

PLOS ONE

Additional Editor Comments (optional):

Reviewers' comments:

Reviewer's Responses to Questions

**Comments to the Author**

1. If the authors have adequately addressed your comments raised in a previous round of review and you feel that this manuscript is now acceptable for publication, you may indicate that here to bypass the “Comments to the Author” section, enter your conflict of interest statement in the “Confidential to Editor” section, and submit your "Accept" recommendation.

Reviewer #1: All comments have been addressed

Reviewer #2: All comments have been addressed

2. Is the manuscript technically sound, and do the data support the conclusions?

Reviewer #1: Yes

Reviewer #2: Yes

3. Has the statistical analysis been performed appropriately and rigorously? 

Reviewer #1: N/A

Reviewer #2: I Don't Know

4. Have the authors made all data underlying the findings in their manuscript fully available?

Reviewer #1: Yes

Reviewer #2: Yes

5. Is the manuscript presented in an intelligible fashion and written in standard English?

Reviewer #1: Yes

Reviewer #2: Yes

6. Review Comments to the Author

Reviewer #1: The authors have addressed the major issues well. They added short subtitles in Methods section, re-organized Table 4 to better link to Figure 2 for cross-reference. They also expanded the Discussion section by highlighting the limitations of the current release and providing recommendations for initial setup and analysis. The minor points were also addressed by adding footnotes and clarifying ambiguities in a few places.

Reviewer #2: The reason I asked "Why don't you use public SMTP server such as gmail to simplify the local installation

and setup of your software?" in the last review is because you mentioned in your manuscript, "Academic Tracker relies on sendmail or an emulator being installed and configured on the machine for its email functionality.". I felt this requirement might limit the use of your software by non technical savvy users because they need to set up sendmail server.

7. PLOS authors have the option to publish the peer review history of their article (what does this mean?). If published, this will include your full peer review and any attached files.

Reviewer #1: No

Reviewer #2: No

---

## [Editor Report · Acceptance letter]

8 Nov 2022

PONE-D-22-09700R1 

Academic Tracker: Software for Tracking and Reporting Publications Associated with Authors and Grants 

Dear Dr. Moseley:

I'm pleased to inform you that your manuscript has been deemed suitable for publication in PLOS ONE. Congratulations! Your manuscript is now with our production department. 

Kind regards, 

on behalf of

Dr. Yuji Zhang 

Academic Editor

PLOS ONE